# Towards Characterization of Hass Avocado Peel and Pulp Proteome during Postharvest Shelf Life

**DOI:** 10.3390/proteomes12040028

**Published:** 2024-09-28

**Authors:** Carolina Camacho-Vázquez, José Miguel Elizalde-Contreras, Francisco Antonio Reyes-Soria, Juan Luis Monribot-Villanueva, José Antonio Guerrero-Analco, Janet Juarez-Escobar, Olinda Velázquez-López, Thuluz Meza-Menchaca, Esaú Bojórquez-Velázquez, Jesús Alejandro Zamora-Briseño, Monica Ramirez-Vazquez, Guadalupe Alheli González Barrenechea, Enrique Ibarra-Laclette, Eliel Ruiz-May

**Affiliations:** 1Red de Estudios Moleculares Avanzados, Instituto de Ecología A. C., Carretera Antigua a Coatepec 351, El Haya, Xalapa 91073, Mexico; carolina.camacho@inecol.mx (C.C.-V.); jose.elizalde@inecol.mx (J.M.E.-C.); antonio.ry.16@gmail.com (F.A.R.-S.); juan.monribot@inecol.mx (J.L.M.-V.); joseantonio.guerrero@inecol.mx (J.A.G.-A.); olinda.velazquez@inecol.mx (O.V.-L.); esau.bojorquez@inecol.mx (E.B.-V.); alejandro.zamora@inecol.mx (J.A.Z.-B.); enrique.ibarra@inecol.mx (E.I.-L.); 2Facultad de Biología, Universidad Veracruzana, Zona Universitaria, Xalapa 91090, Mexico; 3Laboratorio de Genómica Humana, Facultad de Medicina, Universidad Veracruzana, Médicos y Odontólogos S/N, Col. Unidad del Bosque, Xalapa 91010, Mexico; thuluz@gmail.com; 4Unidad de Microscopía, Facultad de Medicina, Universidad Nacional Autónoma de México (UNAM), Edificio “A” PB, Circuito Interior, Avenida Universidad 3000, Ciudad Universitaria, Coyoacán, Ciudad de México 04510, Mexico; 5Centro de Investigación Científica de Yucatán, Carretera Sierra Papacal—Chuburná Puerto, Parque Científico y Tecnológico, Mérida 97302, Mexico; guadalupe.gonzalez@estudiantes.cicy.mx

**Keywords:** Hass avocado, postharvest, proteome, cuticle, flavonoids, lignin

## Abstract

In recent years, avocados have gained worldwide popularity as a nutritive food. This trend is causing a rise in the production of this fruit, which is accompanied by several problems associated with monocultural practices. Despite massive economic gains, limited molecular and structural information has been generated about avocado ripening. In fact, limited studies have attempted to unravel the proteome complexity dynamics of avocado fruit. We therefore conducted a comparative proteomics study on avocado peel and pulp during the postharvest shelf life using tandem mass tag synchronous precursor selection triple-stage mass spectrometry. We identified 3161 and 1128 proteins in the peel and pulp, respectively. Peels exhibited major over-accumulation of proteins associated with water deprivation and oxidative stress, along with abscisic acid biosynthesis. Ethylene, jasmonic acid, phenylpropanoid, and flavonoid biosynthesis pathways were activated. Structurally, we observed the accumulation of lignin and a reduction in cuticular thickness, which coincides with the reduction in the levels of long-chain acyl-coenzyme A synthetase and a marginal increase in 10,16-dihydroxyhexadecanoic acid. Our study sheds light on the association of proteome modulation with the structural features of Hass avocado. Its detailed characterization will provide an alternative for better preservation during the postharvest period.

## 1. Introduction

The avocado (*Persea americana* Mill.) is a popular, nutritious tropical fruit whose origin has been difficult to trace. The avocado was domesticated by the first Mesoamerican cultures, which followed the Mayan and Olmec cultures. Evidence of human selection based on breeding practices suggests that current fruit size was reached between 4000 and 2800 B.C.E. in the Tehuacán and Oaxaan regions [1]. Currently, avocados are mass-produced in Mexico, the USA, Colombia, Indonesia, Chile, the Dominican Republic, Kenya, and South Africa [2].

The Hass avocado cultivar is the most commercialized worldwide and is a Guatemalan × Mexican hybrid generated in California in the 1920s. Its buttery flavor and physical appearance have gained great popularity [3]. However, approximately 30% of this climacteric fruit is lost during the postharvest period due to diseases such as anthracnose and pedicle rot, caused by the fungi *Colletotrichum gloeosporioides* and *Lasiodiplodia theobromae*, respectively [4]. In addition, physiological disorders during postharvest shelf life (PSL), such as ripening heterogeneity, chilling injury, or flesh bruising, reduce the marketable production of this fruit [5].

The ripening process of climacteric fruits involves numerous biochemical changes, including increases in respiration and ethylene production, as well as the softening of the pulp and various metabolic activities that lead to changes in the profiles of carbohydrates, organic acids, and lipids, as well as phenolic and volatile compounds [6]. Structural processes such as cuticle and cell wall biogenesis lead to softening and other quality traits desired by consumers [7]. However, studies examining the cuticular and cell wall tissue during ripening are still scarce in avocado fruits. Moreover, we are far from profiling the full proteome, including proteoform and posttranslational modification in avocado during PSL. This is in part due to a strong research focus over the past decade on the physiological features of avocados under different storage conditions to improve the quality of the final product for human consumption [8].

Proteomics data are essential complementary information to genomics and metabolomics in the quest to carry out a system biology analysis. However, the genomics output regarding the number of gene determinations is significantly higher than the outputs of proteomics and metabolomics in most works related to postharvest fruit biology [9]. Due to the paramount information provided by proteomics under the multiomic approach, it is necessary to improve proteomic protocols in recalcitrant fruit tissues. The complexity of dealing with recalcitrant fruit tissues is exemplified in recent proteomic studies on mango peels, where combining different types of extraction protocols and proteomic workflows allowed the profiling of thousands of proteins in the postharvest period [10]. In addition, previous proteomic analyses of avocados reflected the difficulty of analyzing fruit tissues for which there is limited genomic information [11]. Another barrier in conducting proteomic examinations of avocados is the poor functional annotation of current genomic information, which relies in most cases on homology to plant system models like *Arabidopsis thaliana* and rice (*Oryza sativa*). Recently, a multiomic approach was used to understand the de-synchronization between color and softening during the ripening of Hass avocados under different storage conditions [12], which suggested a tight regulation of key transcription factors linked to the endogenous content of plant growth regulators and flavonoids in the de-synchronized avocado-ripening defect. Although a multiomic approach was utilized in this previous work, transcriptomic data provided more enriched information than proteomic and metabolomic data.

Despite the great commercial interest in avocado, very few proteomic studies have been conducted to understand its PSL. The proteome dynamics and structural characteristics of avocado fruit are closely related to perishability during the postharvest period, as well as susceptibility to and/or tolerance of biotic factors such as anthracnose. Thus, by examining the proteome complexity and structural aspects during fruit ripening, we can better understand the molecular processes involved in cuticle formation and modification during the postharvest period, which is essential for designing appropriate storage conditions [13].

## 2. Materials and Methods

### 2.1. Plant Material

Hass avocados were obtained from an orchard in Huatusco, Veracruz, Mexico (19° 15′88.9″ N, 96° 98′62.0″ W). Green mature Hass avocados were manually harvested based on the best harvest practices of local growers [14]. We collected three avocados from each of three different trees in the same orchard (total of nine fruits per cultivar). Avocados were stored under laboratory conditions at 24 °C and 75% relative humidity after harvest. For this experiment, we sampled fruits at four different postharvest stages: harvest time (T0), three days after harvesting (T3), six days after harvesting (T6), and nine days after harvesting (T9) as previously reported [15]. In each case, three replicates were included, each consisting of three fruits from three different trees. The peel and pulp of each fruit were frozen in liquid nitrogen and stored at −80 °C until analysis.

### 2.2. Determination of Water Transpiration and Firmness

Analyses were carried out at room temperature with three biological replicates. Seven avocados were used from each tree for each measurement time (21 avocados per postharvest stage). The rates of water transpiration were determined by measuring the daily change in weight (g) of avocado fruit as reported previously [16]. The weight loss (WL) was calculated by subtracting the final weight (FW) from the initial weight (IW) of each fruit (IW − FW = WL). This was normalized to the initial weight, yielding the percent water loss (WL − IW/100). The means and standard errors are reported. Avocado firmness was measured using a penetrometer with a 2 mm diameter stainless steel probe (FT011, http://www.qasupplies.com/; 5 February 2021). The firmness was measured at three opposite points along the equatorial zone of the fruit; units for the measurements were Newtons.

### 2.3. Quantification of 1-Aminocyclopropane (ACC) and Abscisic Acid (ABA) by Multiple Reaction Monitoring (MRM)

LC-MS and MRM analysis of growth regulators was carried out as previously described in mango tissues [10]. We pooled the tissues from the three fruits harvested from the same tree, generating three biological replicates. The MRM acquisition method of ACC was performed in an ultra-high-performance liquid chromatograph (UPLC Agilent, 1290, Santa Clara, CA, USA) coupled to a triple quadrupole mass spectrometer (MS Agilent, 6460). The MRM of ABA was carried out in a UPLC (Waters, I-Class) coupled to quadrupole time of flight MS (QTOF, Waters, Synapt G2 Si). Detailed information of the protocol is provided in the Appendix A.

### 2.4. Protein Analysis

A phenol-based protocol was used for protein extraction as reported previously [10]. Samples were digested with trypsin and were used for peptide tagging with Tandem Mass Tag (TMT) 6-plex reagents (Thermo Scientific, Waltham, MA, USA). The T0 samples were tagged in the following order: 126, 127 N, and 128 C. The T3, T6, and T9 samples were tagged with 129 N, 130 C, and 131. Then tagged peptides were fractionated by high-pH reversed-phase liquid chromatography spin columns (Pierce^™^ High pH Reversed-Phase, HPRP). We ended with four fractions with increasing concentrations of acetonitrile (12.5, 15, 17.5, and 20%). All fractions were desalted using ZipTip-C18 tips and dried using a Labconco CentriVapTM vacuum concentrator (Kansas City, MO, USA).

Nano LC-MS/MS analysis was conducted in an Orbitrap Fusion Tribrid (Thermo–Fisher Scientific, San Jose, CA, USA) mass spectrometer equipped with an “EASY spray” nano ion source (Thermo–Fisher Scientific) and interfaced with an UltiMate 3000 RSLC system. The Synchronous precursor selection (SPS)-MS3 protocol was also conducted as reported previously [10]. The MS3 spectra were acquired as previously described by McAlister et al. [17]. The TMT spectra generated by SPS-MS3 were processed with Proteome Discoverer 3.0 (PD, Thermo–Fisher Scientific) and the search engines SEQUEST HT and INFERYS. The resulting peptide hits were filtered for a maximum of 1% False Discovery Rate (FDR) using the Percolator algorithm. We considered at least two peptides, protein coverage equal to or greater than 10%, and a coefficient of variation lower than 30% for accurate protein identification and quantification. Data are available via ProteomeXchange with identifier PXD047227. Proteomics protocols are described in detail in in the Appendix A.

### 2.5. Bioinformatics Analyses

Abundance proteins determined by Inferys were normalized based on the most abundant protein detected through the TMT reporter and precursor ion. We applied the linear model for microarray data (limma)-voom R package V.3.58.0 to visualize the differences in protein abundances between T0, T3, T6, and T9 with a cutoff value of Log2 FC 1.5 (≥0.58, ≤− 0.58) and a *p*-value ≤ 0.05 [18]. Principal component analysis (PCA) and k-means clustering were performed in the cluster R package [19]. Protein abundances were scaled before correlation matrix or k-means analysis, and visual representation was conducted with the ggplot2 R package. For functional annotation, we used the EggNog mapper platform [20]. For this purpose, we uploaded the predicted proteins from a customized avocado database, the output of which was used as the background for gene ontology (GO) enrichment in the ClusterProfiler R package. For GO enrichment of selected proteins we used the following parameters: pvalueCutoff = 0.05, pAdjustMethod = “BH”, qvalueCutoff = 0–0.5, and minGSSize = 2 [21]. The circular network and dot plot (cnetplots) were generated using the Bioconductor R package (version 3.19) for reactome pathway analysis and visualization [22]. For the pathway reconstruction, we used the tool ‘cellular overview’ in the plant metabolic pathway database (PMN) platform (https://www.plantcyc.org/; 5 February 2021) using *Arabidopsis* and *Amborella trichopoda* homologs. The scripts used for this study are presented in the Appendix A.

### 2.6. Bright-Field and Confocal Microscopy

For analysis using bright field microscopy, the pericarp that had previously been cut and stored at −80 °C was fixed for six hours in a solution of formaldehyde in citrate buffer (citric acid 0.1 M, dibasic potassium phosphate 0.2 M, and sucrose 4%), then removed from the fixing solution and kept ultra-frozen (−80 °C) overnight. The fixed and frozen tissue was then cryosectioned by mounting in cryomolds, adding a tissue-freezing medium (Leica Biosystems, Deer Park, IL, USA) at a temperature of −20 °C. Cross-sections were obtained on a LEICA^®^ CM1520 cryostat (8 µm thick). The sections were placed on a coverslip (60 × 24/0.17 mm) used as a support and stored at 4 °C until use. On the same day as cryosectioning, the tissue was stained with a solution of Oil Red Sigma–Aldrich^®^ (Burlington, MA, USA) (0.1% *w*/*v* in 1/1 chloroform/ethanol solution) for 5 min [16]. The excess solution was removed with phosphate-buffered saline (PBS), and they were covered with a coverslip. The treated sections were observed in a Leica DM6000B epifluorescence microscope (LAS AF 3.2.9702 software, 5 megapixel CCD digital camera (DFC450C)), acquiring images at 63×/1.4 NA in bright field and fluorescence. We captured 5 to 40 images of tissues from each sampling time (T0, T3, T6, and T9) for subsequent image selection and analysis. One image for each sampling time was chosen to measure cuticular thickness using the LASX software (version: 3.4.2.18368) and applying deconvolution (refractive Index 1.518, numerical aperture 1.40) and segmentation thresholding (“Reyni Entropy method”, 8-bit image) to the fluorescence images using the software ImageJ/Fiji version 1.54f.

### 2.7. Microscopy Lignin Staining Analyses

Sections were cut by hand and Wiesner staining was performed; this stain specifically detects the cinnamaldehyde groups of lignins [23]. The sections were incubated with 3% phloroglucinol with concentrated HCl (one volume of HCl and two volumes of 3% phloroglucinol) for 10 min. Observations were made with a Nikon SZM1500 stereomicroscope and photographs were taken with a Digital Sight DS-2Mv camera.

## 3. Results

### 3.1. Hass Avocado Ripening during PSL

After harvesting, Hass avocado fruit exhibited a matte green skin color and 81 N of firmness (day one). At three days of PSL, we visualized a slight increment in firmness and similar coloration as on day one (Figure 1B). After six days of PSL, the surface began to change to dark glossy coloration, first around the pedicel, then spreading to the rest of the fruit surface by nine days of PSL (six to nine days, Figure 1A). Firmness began to decrease after three days of PSL and continued to decrease throughout the rest of the analysis period. Loss of weight due to transpiration was constant throughout the PSL (Figure 1B). The determination of the endogenous content of essential plant growth regulators showed an increase in abscisic acid (ABA) after three days of PSL (Figure 1C), while 1-aminocyclopropane-1-carboxylic acid (ACC), the precursor of ethylene, remained nearly constant (Figure 1D).

### 3.2. Peel and Pulp Proteome Landscape

Sodium dodecyl sulfate–polyacrylamide gel electrophoresis (SDS-PAGE) showed similar patterns of protein bands between the peel and pulp samples. The pulp samples had slightly more intense bands than the peel (Figure 2A). We adopted a tandem mass tag synchronous precursor selection triple-stage mass spectrometry (TMT-SPS-MS3) approach, which allowed us to identify more proteins in peel than in pulp samples (Figure 2B).

The list of proteins identified at different time points in peels and pulps are presented in Appendix A, respectively. Overall, we identified 2116 peel-specific proteins, but only 83 pulp-specific proteins (Figure 2C); an additional 1045 proteins were identified in both peel and pulp. Gene ontology (GO) enrichment-based biological processes showed more GO terms in peel than pulp samples, while the biological processes for T6/T0 did not have enriched proteins (Figure 2D). The main biological process was the response to stress in both tissues. Considering a protein count of 30 as the cutoff, most biological processes (e.g., metabolism of amides, organic acids, and xenobiotics) were only visualized in peel tissues. Moreover, additional GO terms associated with stress responses in peels included responses to cold and oxidative stress, which were overrepresented in the last period of PSL.

A more detailed analysis of proteins associated with the response to cold (GO:0009409) identified 66 proteins, among which only a small number were significantly modulated during the last period of PSL (Figure 2E). Acyl-CoA-binding domain-containing protein 2 (ACBP2), alpha-glucan water dikinase 1 (GWD1), cinnamoyl-coenzyme A (CoA) reductase 1 (CCR1), and catalase were over-accumulated, while ribulose-1,5-bisphosphate carboxylase/oxygenase (RuBisCo), plasma membrane-associated cation-binding protein 2 (PCAP2), beta-glucosidase (BGL), and glutathione S-transferase (GST) were under-accumulated (Figure 2E). Several proteins associated with the reaction to cold (GO:0009409) were also related to the response to water deprivation (GO:00094140, Figure 2E). Proteins such as sucrose synthase (SUS1), phospholipase D (PDG), annexin, and S-adenosylmethionine synthase (SAMS) were over-accumulated in the last period of PSL. The plasma membrane-associated cation-binding protein 2 (PCAP2), endoplasmin, and 30S ribosomal protein demonstrated contrasting patterns.

A total of 73 proteins involved in the response to oxidative stress (GO:0006979) were visualized, among which galactinol—sucrose galactosyltransferase (RFS2), stachyose synthase (STSYN), protein transport protein (SEC24), phenylcoumaran benzylic ether reductase (Pyrc5), ascorbate peroxidase 3 (APX3), and protein disulfide isomerase-like 1-4 (PDIL1-4) were over-accumulated. The nicotinamide adenine dinucleotide + hydrogen (NADH)-ubiquinone oxidoreductase 75 kDa subunit, mitochondrial (NDUS1), delta-1-pyrroline-5-carboxylate synthase 1 (P5CS1), NADH dehydrogenase [ubiquinone] flavoprotein 2 (NDUFV2), and glyoxylate/succinic semialdehyde reductase 2 (GLYR2) were under-accumulated (Figure 2F). Several proteins, in addition to their involvement with the reaction to oxidative stress (GO:0006979), were also connected with the response to reactive oxygen species (ROS; GO:0000302) and toxic substances (GO:0009636). For example, 1-aminocyclopropane-1-carboxylate oxidase 2 (ACCO2), inositol-3-phosphate synthase (INO), ferritin-3, and catalase isozyme 1 (CAT-1) were over-accumulated, while nucleoside diphosphate kinase (NDK), thioredoxin X (TRXX), nucleoside diphosphate kinase B (NDKB), and the ionotropic glutamate receptor (nucleoside diphosphate kinase 2, NME2) were under-accumulated.

### 3.3. Pattern of Hass Avocado Proteome Dynamics and Differential Proteins during PSL

Principal component analysis (PCA) showed, in both the peel and pulp samples, that proteins associated with an early time of analysis (T0 and T3) were grouped in the same quadrant (Figure 3A,B). However, peel proteins exhibited more differences between T0 and T3 than pulp proteins, exhibiting almost identical grouping (Figure 3B). Major differences were visualized between early and intermediate (T6) and late (T9) analysis times, suggesting significant modulation of the peel and pulp proteome. We visualized higher cos2 values at T9 in peel but at T0 in pulp, specifying a strong representation in the principal component. These trends of protein accumulation in both peel and pulp tissues were explained by the first two PCA components, which accounted for more than the 90% of the total variation. We then submitted our proteomic data to a k-means clustering, which yielded three main groups based on normalized protein abundances (Figure 3C). This clustering analysis also explained more than 90% of the total variation with its first two components. The first cluster (blue color) corresponded to proteins such as the enolase, fructose-bisphosphate aldolase (FBA), and succinyl-aminoimidazole-carboxamide ribonucleotide synthetase (PUR7), which were determined at a higher proportion in fruit peels than in pulps. Catalase, adenosine triphosphate (ATP) synthase subunit beta (ATP5B), glutathione S-transferase (GST), glyceraldehyde 3-phosphate dehydrogenase (GAPCP) and phosphoglycerate kinase (PGK) followed the same trend. The second cluster (orange) contained aldehyde dehydrogenase (ALDH2B), abscisic stress-ripening protein 2-like (ABA_SRP), 14-3-3-like protein, and polygalacturonase (PG), all with an average protein abundance in peels. The third cluster contained the proteins that were the least abundant in peels, such as endochitinase (CHI), 1-aminocyclopropane-1-carboxylate oxidase (ACCO2), and endoglucanase 1 (GUN).

By focusing particularly on those proteins with a log2 fold change (T3/T0, T6/T0, and T9/0) of >0.58 and <−0.58, and a *p* value of 0.05, we were able to visualize more differential proteins in peel than pulp samples after six and nine days of the postharvest period (Figure 4A). We determined 525 and 405 specific differential proteins in peel and pulp, respectively (Figure 4B), while a total of 191 proteins were determined as differential in both peel and pulp samples. Gene ontology enrichment showed biological processes such as a response to stress, generation of precursor metabolites and energy, monocarboxylic, nucleobase-containing small molecule, and xenobiotic metabolisms, which were overrepresented in both peel and pulp samples (Figure 4C). Meanwhile, the biological processes in T3/T0 peel and T6/T0 pulp did not have enriched proteins (Figure 4C). In contrast, carbohydrate-derivative biosynthesis, the defense response to bacteria, photosynthesis, cold, oxidative stress, and toxic substances, was overrepresented in the peel. Due to the lower representation of secreted protein in GO enrichment protocols, we predicted the occurrence of signal peptide (SP) in all determined differential proteins. By doing so, we were able to visualize a small proportion of proteins with a positive prediction of SP (9%). Peel proteins demonstrated a more dynamic accumulation of the predicted secreted proteins (Figure 4E). For example, endochitinase 2 (CHI2), pathogenesis-related proteins (PR1B, PR-4B), thaumatin-like protein (TLP), non-specific lipid-transfer protein Lac s 1 (lipid transfer protein, LTP), probable pectate lyase 1 (PL1), and polygalacturonase (PG) were over-accumulated during the postharvest period. The endoplasmin homolog (Enp), aspartyl protease family protein (APC), rhamnose:rhamnosyltransferase 1 (URT1), peroxidase 4 (POX4), expansin-A4 (EXPA4), alpha-xylosidase 1 (XYL1), and subtilisin-like protease (SBT3.17) were under-accumulated.

### 3.4. Modulation of Metabolic Pathways in Avocado Peel and Pulp Tissues

The proteomic data showed the activation of plant growth regulator pathways including ethylene (ET) and jasmonic acid (JA). For example, 1-aminocyclopropane-1-carboxylate oxidase 4 (ACO4) was significantly over-accumulated in both peels and pulp after nine days of PSL, exhibiting higher values in pulp (Figure 5). Proteins related to JA, such as OPC-8:0 CoA ligase (OPCL), abnormal inflorescence meristem 1 (AIM1), and 3-ketoacyl-CoA thiolase (KAT2), were mostly over-accumulated in peel. UDP-glycosyltransferase 85A1 (UGT85A1), associated with the cytokinin-O-glycosylation, followed the same trend as JA-related proteins. The general activation of ET and JA was correlated with that of the phenylpropanoid pathway, more in peel than in pulp, during PSL. The most representative enzymes included phenylalanine ammonia-lyase (PAL 2, 4), 4-coumarate--CoA ligase (4CL7), cinnamoyl-CoA reductase (CCR1), eugenol, and isoeugenol synthase. A similar activation of the flavonoid biosynthetic pathway was observed in peels. Our proteomic data demonstrated the overaccumulation of naringenin chalcone synthase (CHS) and (S)-naringenin 3-dioxygenase (F3H2), enzymes associated with the methylation of quercetin and chrysoeriol biosynthesis such as O-methyltransferase 1 (OMT1), coniferyl alcohol dehydrogenase (CAD4), and caffeoyl-CoA O-methyltransferase (CCoAOMT), which were over-accumulated during PSL.

### 3.5. Structural Behavior of the Cuticle during PSL

Due to the importance of cuticle structure during climacteric fruit ripening, we attempted to identify proteins associated with cutin biosynthesis. It was only possible to visualize an under-accumulation of long-chain acyl-CoA synthetase 1 (LACS1) (Figure 6A). A cross-section of Hass avocado pericarp exhibited a well-defined cuticle deposition with pronounced anticlinal pegs after fruit collection (Figure 6B–D). During the postharvest period, there was a reduction in the thickness of the cuticle. Significant differences in cuticle thickness were detected between fruits after harvesting and after nine days of PSL (Figure 6B–D). Moreover, during the postharvest period, some epidermal cells were completely covered with cutin. In addition to the aforementioned cuticular changes, after fruit collection, lignin deposition was visible in regions of the hypodermis-based phloroglucinol–HCl staining (Figure 6D, T0). Lignin deposition increased constantly and substantially during PSL, reaching regions of the epidermis (Figure 6D, T3–T9). The phloroglucinol–HCl staining also changed the coloration of cuticles after six and nine days of PSL, suggesting modifications in the chemical profile of the cuticle constituents (Figure 6D). The cutin depolymerization showed four major peaks, most of which had similar proportions throughout the PSL, while the one associated with 10,16-dihydroxyhexadecanoic acid (10,16-diOH) was slightly over-accumulated after nine days.

## 4. Discussion

Avocado is a highly appreciated fruit for its organoleptic characteristics, with Mexico being the main producing and exporting country. The global demand for avocado by 2030 is estimated to reach 2.14 million tons and a value of USD 4,655 million [24]. During PSL, avocado fruits undergo important structural and biochemical changes from harvest maturity to ripening; however, the detailed physiological and proteome modulations of avocados at PSL under normal conditions are unknown for most cultivars.

### 4.1. Stress Response Was a Main Factor in Hass Avocado Proteome Modulation during Postharvest Shelf Life

The ripening of the avocados is delayed by several months while on the trees due to the endogenous content of mannoheptulose and perseitol, whose variation throughout the tree results in asynchronous ripening within trees [25]. Based on horticultural practices, avocados intended for distant destinations are stored in cold and controlled atmosphere conditions, with the main goal of inhibiting ripening to maintain the highest possible quality of the final product [26].

This proteomic study highlights the stress response as a major enriched biological process (Figure 2). It is worth noting the over-accumulation of proteins associated with the response to cold, water transpiration, and oxidative stress, such as alpha-glucan water dikinase 1 (GWD1), cinnamoyl-CoA reductase 1 (CCR1), catalase, sucrose synthase (SUS1), phospholipase D (PDG), stachyose synthase (STSYN), ascorbate peroxidase 3 (APX3), protein disulfide isomerase-like 1-4 (PDIL1-4), and 1-aminocyclopropane-1-carboxylate oxidase 2 (ACCO2, Figure 3). The positive modulation of these proteins is directly associated with the significant amount of water transpiration and over-accumulation of the endogenous content of ABA of Hass avocados during PSL (Figure 1B,C). A similar trend of ABA accumulation was observed in the Criollo and Kent mango cultivars, which is also related to cuticular features such as cuticle thickness, intra- and epicuticular wax deposition, and the occurrence of natural openings such as microcracks and lenticels [16]. Although lenticel damage is a major problem in avocado fruit stored in cold, humid conditions, limited information on the biogenesis and structural features of avocado lenticels has been reported in past decades [27]. Methylcyclopropene (an ethylene inhibitor) and wax treatments delay and improve the PSL of the Tower II and Boot 7 avocado cultivars, suggesting that more structural examinations should focus on natural openings in the avocado fruit surface [28].

Taking a closer view of the over-accumulated secreted proteins determined in our study, pathogenesis-related proteins (CHI2, PR1B, PR-4B, TLP) as well as glucan endo-1,3-beta-glucosidase (BG), non-specific lipid-transfer protein Lac s 1 (LTP), probable pectate lyase 1 (PL1), and polygalacturonase (PG) also exhibited major modulation in the pathogen defense response, in addition to cell wall structural modulation (Figure 4D). The TLP protein family is a complex one associated with host defense and development in plants, whose inductions are visualized under drought and osmotic stresses. Previously, TLP extracted from banana fruit tissues showed the ability to disrupt the plasma membrane and cell wall of *Penicillium expansum* [29], suggesting the occurrence of potential bioactive molecules in avocado fruit tissues for biotechnological applications. In addition, the physiological process associated with cuticular features as well as the intimate relationship between the cuticle and cell wall suggests a tight regulation of the modulation of these two structural features. In this context, the biochemical characterization of cell wall proteins such as PL1 and PG, which play essential roles in fruit softening [30], could provide some alternatives for improving the PSL of avocados.

### 4.2. Secondary Metabolism Active in Avocado Peel and Pulp Proteome during Fruit Ripening

The ethylene and jasmonic acid metabolic pathways were active during the PSL of Hass avocados, along with the phenylpropanoid metabolic pathway (PPMP), which represents the main gateway to producing a vast number of secondary metabolites in plants (Figure 5). In general, the PPMP is activated during different types of abiotic stress, and the PSL is no exception, with the purpose of coping with environmental constraints [31]. Gene ontology enrichment connects water transpiration with the response to water deprivation and oxidative stress during PSL (Figure 3). This biological process modulates the essential gene-encoding enzymes of PPMP, resulting in the increase in endogenous content of phenolic-related molecules. The nature of these phenolic compounds ranges from lignin monomers to flavonoids, anthocyanin, and coumarins.

Our data suggested the activation of the accumulation of lignin, with the overaccumulation of key enzymes such as CCR1, CAD4, OMT1, eugenol synthase (EGS), and isoeugenol synthase (IGS) during PSL (Figure 5). Hass avocados exhibited an increase in lignin during development until reaching physiological maturity [32]. However, lignification due to external signals during PSL contributed to the deterioration of fruit quality, as has been observed in other climacteric fruits [33]. In addition, the lignification of the peels inhibited the cell wall breakdowns during ripening, thus altering the proper softening of fruits and flavor, and decreasing the juice yield, all of which affect desirable traits for consumers [34]. Our proteomic data correspond with previous studies related to the accumulation of lignin during PSL of Hass avocados. Moreover, phloroglucinol–HCl staining showed a significant accumulation of lignin during PSL (Figure 6D). Therefore, our paper provides useful information for designing postharvest treatment and storage conditions to regulate the levels of lignin production. Recently, the application of melatonin to water bamboo shoots led to a delay in lignification, improving the quality of this vegetable and several fruits [35].

Previously, a metabolomic study of avocado peels showed the occurrence of organic and phenolic acids, flavonoids, catechins, procyanidins, phenylpropanoids, lignans, sugars, fatty acids, and other polar compounds [36]. Recently, anthocyanins, catechins, and epicatechins along with ABA and brassinosteroids were associated with ripened color-softening synchronized black avocados [12]. In our proteomic study, we provide data that demonstrate the activation of the flavonoid biosynthetic pathway, determining the essential contribution of CHS, 4CL7, F3H2, and CCoAOMT during PSL (Figure 5). Flavonoids, due to their hydroxyl groups, play an essential role in maintaining redox homeostasis by counteracting reactive oxygen species (ROS) [37]. Therefore, an active production of flavonoids could provide better conditions during PSL by avoiding membrane deterioration and lipid peroxidation as a consequence of the burst of ROS. It is worth noting that the recalcitrancy of Hass avocado fruit tissues and poor genome functional annotation are the main limitation for the complete profiling of flavonoid-related enzymes. Alternative proteomic pipelines such as label-free or data-independent acquisition approaches and the newest mass spectrometers will provide better proteome information.

### 4.3. Hass Avocado Cuticle Structural Modifications during PSL

The cuticle is composed of waxes, polyphenolic compounds, and cutin that contribute to fruit development and the modulation of the postharvest shelf life, as has been well documented in several species [7]. Cuticular features can regulate water loss, softening, and resistance to diseases and insect pests, and subsequently, postharvest fruit quality [13]. Cuticle chemical composition and morphology vary among cultivars within the same species. For example, comparative analysis between harvest time and after the controlled atmosphere storage of two apple varieties—Szampion and Jonagold—revealed major differences in cuticular thickness and waxes [38]. The Jonagold variety possessed a thicker cuticle than the Szampion after harvest and storage time. In addition, crystalline wax was more abundant on the Jonagold fruits after both the harvest and storage time, which was associated with a lower loss of weight from water transpiration. In a later study on mango, the premium cultivar Tommy Atkins presented a thickened cuticle, abundant epicuticular waxes, and fewer lenticels and microcracks than Criollo cultivars during PSL, which was associated with lower water transpiration, better firmness, and greater resistance against fruit flies [16]. In October Sun peach fruits, the content of cuticle per surface area (g m^−2^) increased significantly after cold storage, while the waxes did not exhibit different values between the control and cold-treated specimens [39].

In our research with Hass avocados, we observed a reduction in cuticle deposition during PSL along with cutin accumulation below the epidermal cells (Figure 6B,C). This cuticular modification was correlated with the deaccumulation of LACS1, as well as with the slight increase of 10,16-dihydroxyhexadecanoic acid (10,16-diOH) as the main cutin monomer (Figure 6E). The 10,16-diOH has been determined as the main cutin monomer in berries, peppers, sweet cherries, tomatoes, and mangoes [7]. The increase in this cutin monomer could be related to the determination of cutin in deeper areas of the epidermal cells of Hass avocados or to the extractability of cutin monomers due to cuticle structure modification during PSL. It is worth noting that cuticle thickness is not the only feature associated with the level of water transpiration; several other factors include cutin chemical composition, intra- and epicuticular wax content composition, and natural openings like lenticels and microcracks. Therefore, the generation of more robust proteomic information such as the scrutiny of the glycoproteome, an area tightly associated with structural changes in fruits, along with functional inspection will provide more information related to the cuticular features of avocados, with the main goal of designing better conditions for their PSL.

## 5. Conclusions

The comparative proteomic scrutiny based on SPS-MS3 in Hass avocado peel and pulp exhibited a response to cold, water transpiration, and oxidative stress as main molecular features during PSL. The proteome dynamics visualized in our study related to a reduction in cuticle thickness and a significant accumulation of lignin at the last stage of the study. Our scrutiny of avocados underpinned that structural features are an essential factor during PSL. Therefore, integrating detailed proteomic studies such as the profiling of posttranslational modification with other -omics tools will provide additional clues about the molecular foundation of avocado ripening.

## Figures and Tables

**Figure 1 proteomes-12-00028-f001:**
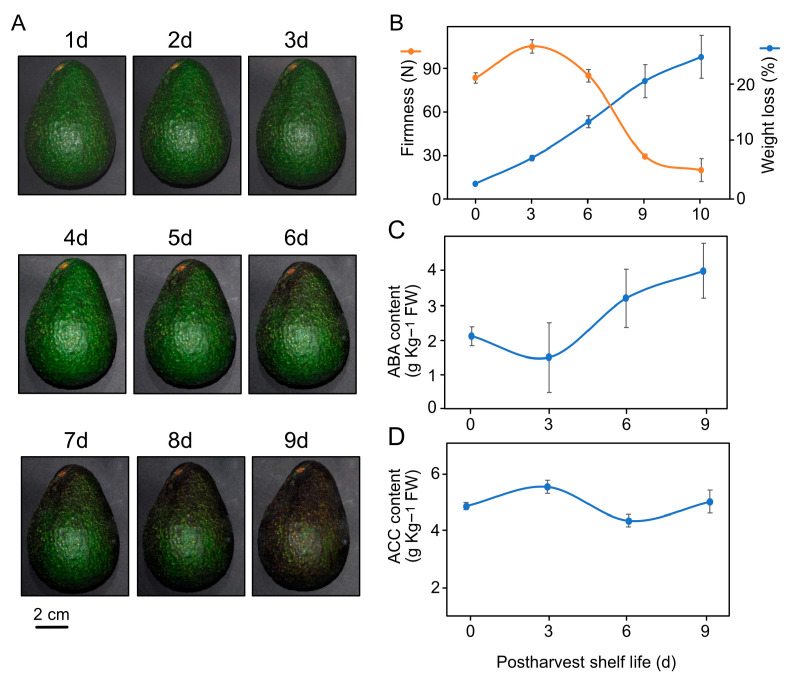
Hass avocado ripening under normal conditions at room temperature. (**A**) Avocado coloration during postharvest shelf life (PSL) (1–9 days). (**B**) Drastic water transpiration and decreased firmness. Endogenous content of (**C**) abscisic acid (ABA) and (**D**) 1-aminocyclopropane-1-carboxylate (ACC) in avocado pericarp in postharvest period.

**Figure 2 proteomes-12-00028-f002:**
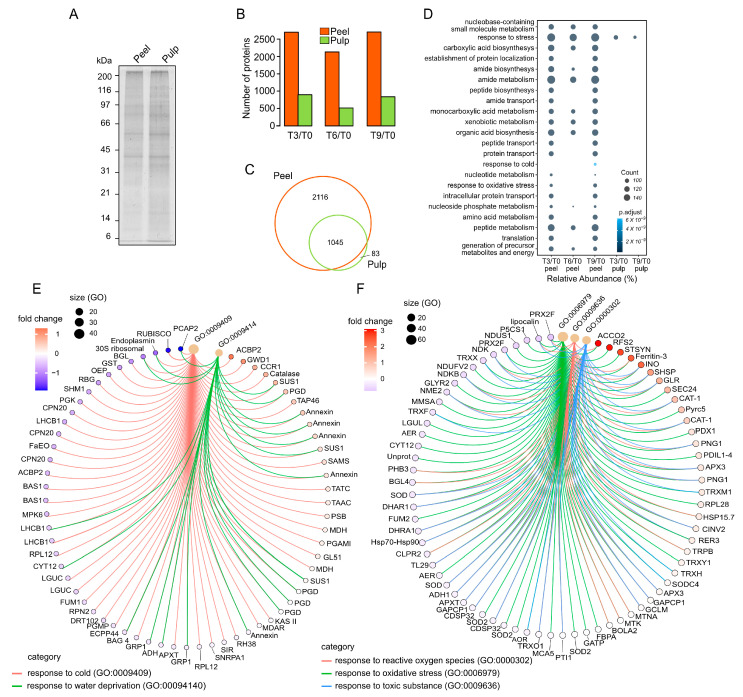
Overview of peel and pulp proteome landscape of Hass avocado during PSL. (**A**) Representative peel and pulp protein band in SDS-PAGE gel. (**B**) Number of proteins and (**C**) grouping identified in peel and pulp by the TMT-SPS-MS3 approach. (**D**) Dot plots considering more than 30 protein-based gene ontology (GO) biological process enrichment analysis. (**E**) Visual representation of circular network of proteins associated with cold and response to water deprivation. (**F**) Visual representation of circular network of proteins associated with response to ROS, oxidative stress, and toxic substance.

**Figure 3 proteomes-12-00028-f003:**
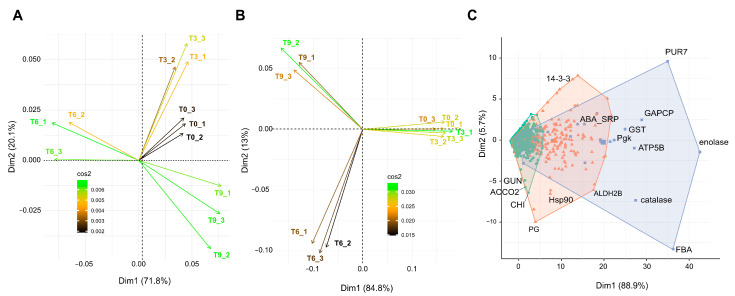
Protein dynamic accumulation in Hass avocado fruit tissues during PSL-based principal component and clustering analyses. Protein abundances identified in PSL period in (**A**) peel and (**B**) pulp samples were normalized by scaling (range 0–1) for PCA. A high cos2 indicates a strong representation of the variable on the principal component. T0 indicates samples just after harvest; T3, T6, and T9 indicate samples after three, six, and nine days of postharvest shelf life. Numbers 1, 2, and 3 in labels specify the three biological replicates. (**C**) k-means clustering of normalized protein abundances in PSL. Phosphoglycerate kinase (Pgk), glutathione S-transferase (GST), glyceraldehyde 3-phosphate dehydrogenase (GAPCP), succino-aminoimidazole-carboximide RN synthetase (PUR7), fructose-bisphosphate aldolase (FBA), ATP synthase subunit beta (ATP5B), abscisic stress-ripening protein 2-like (ABA_SRP), aldehyde dehydrogenase (ALDH2B), polygalacturonase (PG), endochitinase (CHI), 1-aminocyclopropane-1-carboxylate oxidase (ACCO2), Endoglucanase 1 (GUN).

**Figure 4 proteomes-12-00028-f004:**
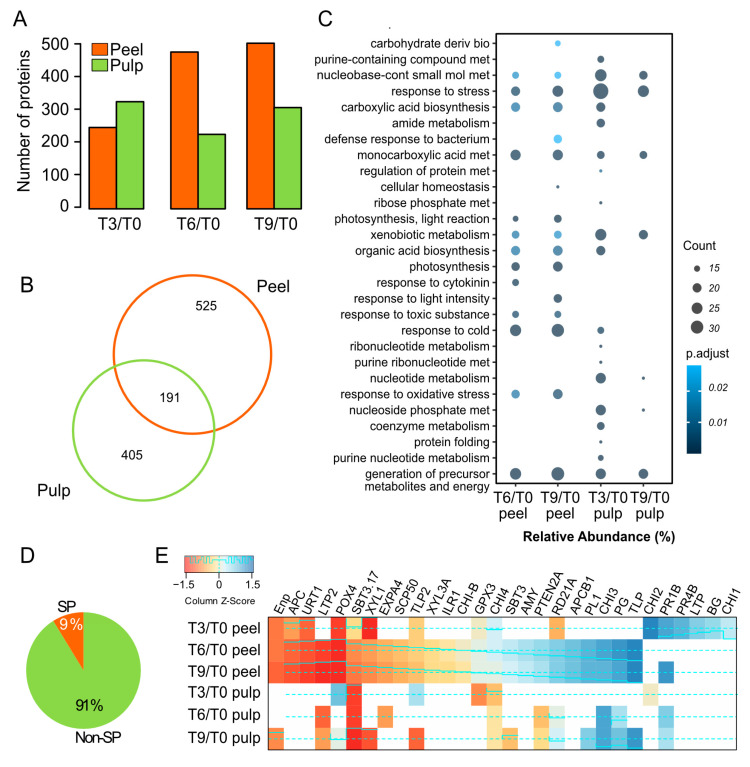
Dynamic of differential proteins during PSL in Hass avocado fruits. (**A**,**B**) Number of differential proteins identified in fruit tissues during PSL. (**C**) Dot plots considering more than 10 protein-based GO biological processes enrichment analysis. (**D**) Percentage of predicted proteins with N-terminal signal peptide (SP) among differential proteins determined in PSL. (**E**) Heatmap of differential protein-predicted with SP. Density and trace lines are displayed in cyan, using both solid and dashed lines to represent the frequency of values in the data. Endochitinase 1 (CHI1), glucan endo-1,3-beta-glucosidase (BG), endochitinase 2 (CHI2), endochitinase 3 (CHI3), probable pectate lyase 1 (PL1), aspartyl protease family protein 1 (APCB1), cysteine proteinase (RD21A), phosphatidylinositol 3,4,5-trisphosphate 3-phosphatase and protein-tyrosine-phosphatase (PTEN2A), alpha-amylase (AMY), subtilisin-like protease (SBT3), endochitinase 4 (CHI4), probable glutathione peroxidase 3 (GPX3), basic form of pathogenesis-related protein 1 (CHI-B), IAA-amino acid hydrolase ILR1-like 6 (ILR1), beta-xylosidase/alpha-L-arabinofuranosidase (XYL3A), thaumatin-like protein (TLP2), serine carboxypeptidase-like 50 (SCP50), alpha-xylosidase 1 (XYL1), non-specific lipid-transfer protein Lac s 2 (LTP2).

**Figure 5 proteomes-12-00028-f005:**
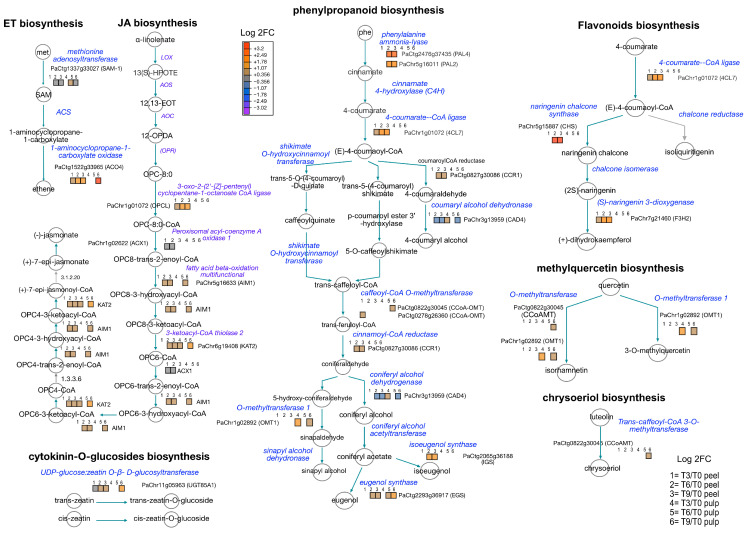
Modulation of metabolic pathways during Hass avocado ripening during PSL. Reconstruction of ethylene (ET), jasmonic acid (JA), phenylpropanoid, and flavonoid biosynthetic pathways based on log2 fold change of differential values. The pathway reconstruction was performed using the tool ‘cellular overview’ in the plant metabolic pathway databased (PMN) platform. We used *Arabidopsis* and *Amborella trichopoda* homologs.

**Figure 6 proteomes-12-00028-f006:**
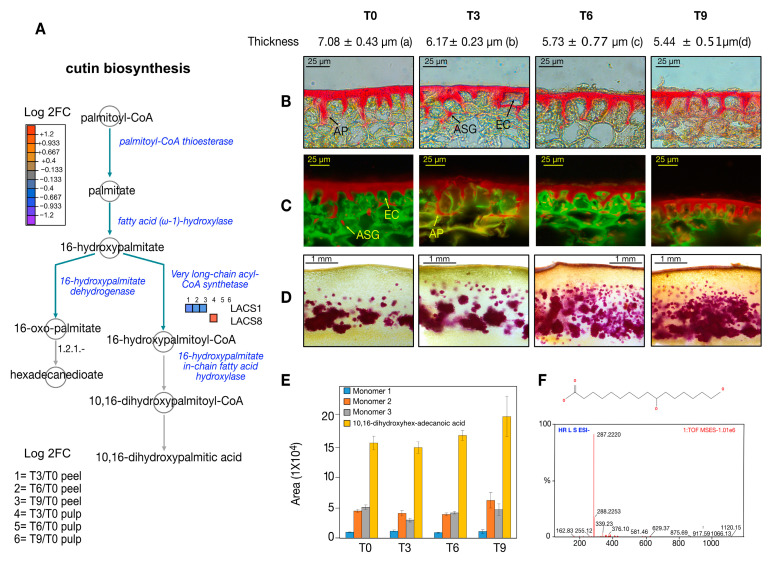
Avocado structural feature modulation during PSL. (**A**) Cutin biosynthesis pathway-based plant metabolic pathway database (PMN) platform. Cuticular morphological features during PSL visualized with (**B**) epifluorescence and (**C**) fluorescence microscopy. (**D**) Lignin visualization using phloroglucinol–HCl staining in pericarp cross-sections. (**E**) Cutin monomer accumulation during PSL and (**F**) 10,16-dihydroxyhex-adecanoic acid MS/MS spectra and assignation by accurate mass spectrometry. In (**B**,**C**) AP, anticlinal pegs; EC, epidermal cells; ASG, attached sub-epidermal globules.

## Data Availability

Data are available via ProteomeXchange with identifier PXD047227.

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
