# Peer review of "Towards Characterization of Hass Avocado Peel and Pulp Proteome during Postharvest Shelf Life"

_proteomes, 2024, doi:10.3390/proteomes12040028_

Round 1
Reviewer 1 Report
Comments and Suggestions for Authors
Authors have characterized proteomic profiles of avocado peel and pulp to understand their post harvest shelf life. The manuscript can improved with few changes.
1) Determination of total protein concentration at different time points.
2) Authors should describe the rationale of choosing different time points
3) Authors should provide the list of proteins identified at different time points.
4) Bioinformatics methods should be elaborated, like how pathway reconstruction was performed.
5) Differences between peel and pulp characteristics based on PCA ( Figure 3a and 3b) should be properly explained
Comments on the Quality of English LanguageMinors English correction to be made.
Author Response
1) Determination of total protein concentration at different time points.
Response: Dear reviewer, we added the total protein concentration at different point in supporting information. In general, we visualized the same content in our extractions.
Table 1. Protein concentration of each sample analyzed in our study
|
Peel (μg/µl) |
SD |
Pulp (μg/µl) |
SD |
|
|
T0 |
6.15411826 |
0.55548205 |
7.67272878 |
1.36055495 |
|
T3 |
7.56267997 |
1.33065286 |
7.3978575 |
1.82553967 |
|
T6 |
6.9490989 |
1.51590166 |
5.76856004 |
1.13197914 |
|
T9 |
6.9788874 |
1.53685399 |
6.41234845 |
1.44893215 |
2) Authors should describe the rationale of choosing different time points
Response In previous proteomic studies on avocados (https://doi.org/10.1016/j.jprot.2021.104112), we visualized the Hass avocado ripened around ten days of postharvest shelf life. In this study, we focused on seed proteomics. However, physiological parameters (phenotypical and appearance features, weight loss, and firmness) were determined in the whole fruit. Therefore, this second study focused on the peel and pulp of Hass avocado; we also conducted our analysis within the ten days of postharvest shelf life and selected T0, T3, T6, and T9 for proteomic scrutiny.
We added in line 100: ‘as previously reported [14]’
3) Authors should provide the list of proteins identified at different time points.
Response: Dear reviewer many thanks for this essential observation, we added in lines 208-209 the following specifications ‘The list of proteins identified at different time points in peels and pulps are presented in Table S1 and S2, respectively’
4) Bioinformatics methods should be elaborated, like how pathway reconstruction was performed.
Response: Dear reviewer, we elaborate in this section and we are adding the scripts used for this study in Supplementary Materials.
Abundance proteins determined by Inferys were normalized based on the most abundant protein detected through the TMT reporter and precursor ion. We applied the linear model for microarray data (limma)-voom R package V.3.58.0 to visualize the differences in protein abundances between T0, T3, T6, and T9 with a cutoff value of Log2 FC 1.5 (≥ 0.58, ≤ 0.58) and a p-value ≤ 0.05 [17]. The principal component analysis (PCA) and k-means clustering was performed based in cluster R package [18]. Protein abundances were scaled before correlation matrix or k-means analysis, and visual representation was conducted with ggplot2 R package. For functional annotation we used the EggNog mapper platform [19]. For this purpose, we uploaded the predicted proteins from a customized avocado database, which output was used as the background for gene ontology (GO) enrichment based the ClusterProfiler R package. For GO enrichment of selected proteins we used the following parameters: pvalueCutoff = 0.05, pAdjustMethod= “BH”, qvalueCutoff=0-05 and minGSSize=2 [20]. The circular network and dot plot (cnetplots) were generated using the R/Bioconductor package for reactome pathway analysis and visualization [21]. For the pathway reconstruction, we used the tool ‘cellular overview’ in the plant metabolic pathway database (PMN) platform (https://www.plantcyc.org/) using Arabidopsis thaliana and Amborella trichopoda homologs. The scripts used for this study are presented in Supplementary Materials.
5) Differences between peel and pulp characteristics based on PCA ( Figure 3a and 3b) should be properly explained
Response: Many thanks for this observation, we improved the suggested section as follows:
Principal component analysis (PCA) showed, in both the peel and pulp samples, that proteins associated with an early time of analysis (T0 and T3) were grouped in the same quadrant (Fig. 3A, B). However, peel proteins exhibited more differences between T0 and T3 than pulp proteins, exhibiting almost identical aggrupation (Fig. 3B). Major differences were visualized between early and intermediate (T6) and late (T9) analysis times, suggesting significant modulation of the peel and pulp proteome. In peel, we visualized higher cos2 values at T9 while in pulps at T0, specifying a strong representation in the principal component. These trends of protein accumulation in both peel and pulp tissues were explained by the first two PCA components, which accounted for more than the 90% of the total variation
Reviewer 2 Report
Comments and Suggestions for Authors
Towards characterization of Hass Avocado Peel and Pulp Proteome During Postharvest Shelf Life Review
Comments for the Authors
This article describes comparative analysis of Hass avocado peel and pulp following harvest to fill a gap in knowledge on the details of avocado ripening at the molecular level. The authors have presented data on structural changes including color, water loss, and firmness, and related those back to their analysis of ABA and a detailed proteomic comparison of pulp vs peel over the PSL. The authors relate the water loss and increase in ABA content over the PSL back to over- and under-accumulated proteins including multiple detailed metabolic pathways (including response to cold, oxidative stress, and water deprivation) with quantitative data from the TMT proteomic experiments. Comparison of peel and pulp proteomes showed more unique proteins in peel and larger changes in peel proteome over PSL. The authors go into detail regarding the overall metabolic changes within the proteome over the PSL highlighting certain enriched GO term pathways. Finally, the authors use numerous microscopy analyses to compare the structural feature modifications in the cuticle over the PSL and relate these changes back to their proteomics data.
This manuscript requires minor revisions as detailed below. The experiments as described are adequate to justify the conclusions. In this reviewer’s opinion, this article is of broad interest to Proteomes readership. The authors have described and presented compelling experiments with exceptional Figures that tell a complete scientific narrative.
Recommendation to editor: Accept with minor revisions.
Minor Points:
1. Line 31: “. Structurally, we observed the accumulation of lignin and the reduction”
2. Line 33:
3. Line 55: The reference style is different here as well as a few others. The authors should carefully review the reference formatting.
4. Line 58: “Besides, we are far to profile the proteome complexity including proteoform and posttranslational modification in avocado during PSL.” I believe there was a typo in this sentence and I cannot understand what is the intended meaning here maybe yet instead of far?
5. Line 64: “However, the output in terms of the number of gene profiles is not comparable to proteomic or metabolomic approaches in most works related to postharvest fruit biology” IS this referring to the output of genomics and metabolomics? The authors should revise this sentence to clarify.
6. Line 68: “The complexity of dealing with fruit difficult” should be revised to “The complexity of dealing with difficult fruit”
7. Line 98: “we sampled fruits at three different postharvest stages” should be revised to “we sampled fruits at four different stages”
8. Line 106 (and Figure 1B): How was the water loss calculated? Was it relative to total fruit mass or somehow normalized?
9. Line 125 (and others): No need to include the trademark of Thermo Scientific. If it is included, it should be properly superscripted.
10. Line 134 (and others): Again, no need to include the copyright symbol (©).
11. Line 169: Another differently formatted reference.
12. Line 182: I do not believe the catalogue number is required here.
13. Line 188: “After harvesting, the Hass avocado fruit exhibited a matte green skin color and 27 N of firmness (one day one).” This sentence seems to be at odds with Figure Fig. 1B which suggests that the firmness measurement at day one should be on the order of 90 N. The authors also did not comment on the increase in firmness from harvest to 3d PSL. Perhaps this could be introduced and related to proteome changes between 0 and 3d PSL later in the paper?
14. Paragraph beginning on Line 216: The authors do not discuss the most significantly over-accumulated protein ACBP2 (from Figure 2E). Was this intentionally omitted or an oversight?
15. Line 231: “Number of proteins identified in (B) peel and (D) pulp by the TMT-SPS-MS3 approach” changed to “(B) Number of proteins and (C) grouping identified in peel and pulp by the TMT-SPS-MS3 approach” Perhaps rewrite this sentence or split into two to make it clear what the two subfigures are showing.
16. Line 274: Is SPL a typo of PSL?
17. Line 296: “Due to the lower representation of secreted protein in GO enrichment protocols, we predicted the occurrence of signal peptide (SP) in all determined differential proteins. By doing so we were able to visualize a small proportion of proteins with a positive prediction of SP (9%).” Could the authors better describe this process? I am not understanding what was predicted or how “positive prediction” was visualized.
18. Figure 4B: Is this Venn diagram for all samples or a specific PSL comparison?
19. Line 311: (D) should be changed to (E). A description of subfigure 4D is missing.
20. Figure 6B and C: AP, EC and ASG are not defined. AP = anticlinal pegs, EC = epidermal cells, ASG = ?
21. Line 373: Another reference formatting
22. Line 505: Remove the author instructions
23. SI: There are some formatting errors with units in the SI – specifically there are numerous instances were the μM are missing the “μ”
Major Points:
1. The authors should add a short sentence or two at the end of the abstract to describe the outcome of this study – specifically that their data fills a gap in knowledge regarding avocado PSL and the future of the work.
2. Figure 2D, E and F: The legend scale on all three subfigures should include smaller dots that are closer to the sizes of the dots in the figures. As it is currently presented, readers are required to extrapolate dot sizes smaller to estimate size and count. The legend should span the range of dot sizes included in the figure. This same correction is required for Fig. 4C.
Comments on the Quality of English LanguageExcept for the few minor corrections above regarding unclarity of specific sentences, the English is exceptional / native speaker level.
Author Response
Minor Points:
- Line 31: “. Structurally, we observed the accumulation of lignin and the reduction”
Response: done
- Line 33:
- Line 55: The reference style is different here as well as a few others. The authors should carefully review the reference formatting.
Response: Done
- Line 58: “Besides, we are far to profile the proteome complexity including proteoform and posttranslational modification in avocado during PSL.” I believe there was a typo in this sentence and I cannot understand what is the intended meaning here maybe yet instead of far?
Response: We modified the sentence as follows : ‘Besides, we are far from profiling the full proteome, including proteoform and post-translational modification in avocado during PSL’
- Line 64: “However, the output in terms of the number of gene profiles is not comparable to proteomic or metabolomic approaches in most works related to postharvest fruit biology” IS this referring to the output of genomics and metabolomics? The authors should revise this sentence to clarify.
Response: We modified the sentence as follows: ‘However, the genomic output regarding the number of gene determinations is significantly higher than the outputs of proteomics and metabolomics in most works related to postharvest fruit biology.
- Line 68: “The complexity of dealing with fruit difficult” should be revised to “The complexity of dealing with difficult fruit”
Response: We modified the sentence. ‘The complexity of dealing with recalcitrant fruit tissues’
- Line 98: “we sampled fruits at three different postharvest stages” should be revised to “we sampled fruits at four different stages”
Response: Many thanks for this observation. We fixed this mistake, “we sampled fruits at four different stages”
- Line 106 (and Figure 1B): How was the water loss calculated? Was it relative to total fruit mass or somehow normalized?
Response: Dear review, we added the following lines in the material and methods section, lines 109-111.
“The weight loss (WL) was calculated by subtracting the final weight (FW) from the initial weight (IW) of each fruit (IW−FW=WL). This was normalized to the initial weight, yielding the percent water loss (WL−IW/100)”
- Line 125 (and others): No need to include the trademark of Thermo Scientific. If it is included, it should be properly superscripted.
Response: Done
- Line 134 (and others): Again, no need to include the copyright symbol (©).
Response: Done
- Line 169: Another differently formatted reference.
Response: Done
- Line 182: I do not believe the catalogue number is required here.
Response: Done
- Line 188: “After harvesting, the Hass avocado fruit exhibited a matte green skin color and 27 N of firmness (one day one).” This sentence seems to be at odds with Figure Fig. 1B which suggests that the firmness measurement at day one should be on the order of 90 N. The authors also did not comment on the increase in firmness from harvest to 3d PSL. Perhaps this could be introduced and related to proteome changes between 0 and 3d PSL later in the paper?
Response: Dear reviewer many thanks for visualizing this mistake. We fixed the paragraph now in lines 1997-199. The physiological and structural changes mirror the proteome modulation during the postharvest period. In this manuscript, we focused on those evident cuticle changes and lining changes. It would be great to scrutinize in detail the early stages of PSL, we are sure, we are going to find quite interesting molecular features
After harvesting, the Hass avocado fruit exhibited a matte green skin color and 81 N of “firmness (day one). At three days of PSL, we visualized a slight increment of firmness and similar coloration as on day one (Fig. 1B)”
- Paragraph beginning on Line 216: The authors do not discuss the most significantly over-accumulated protein ACBP2 (from Figure 2E). Was this intentionally omitted or an oversight?
Response: Dear reviewer, we added in the list of over-accumulate proteins specified in Fig 2E. the Acyl-CoA-binding domain-containing protein 2 (ACBP2), now in line 231.
It seems that ACBP2 in Arabidopsis thaliana could relate to cuticle formation and plant responses to microbes (https://www.ncbi.nlm.nih.gov/pmc/articles/PMC3465942/). Although avocado fruit differs entirely from Arabidopsis, we can obtain additional information about cuticle modulation during PSL in avocados.
- Line 231: “Number of proteins identified in (B) peel and (D) pulp by the TMT-SPS-MS3 approach” changed to “(B) Number of proteins and (C) grouping identified in peel and pulp by the TMT-SPS-MS3 approach” Perhaps rewrite this sentence or split into two to make it clear what the two subfigures are showing.
Response: We modified the sentence to “(B) Number of proteins and (C) grouping identified in peel and pulp by the TMT-SPS-MS3 approach”, now in lines 244-245
- Line 274: Is SPL a typo of PSL?
Response: SPL is a typo of PSL
- Line 296: “Due to the lower representation of secreted protein in GO enrichment protocols, we predicted the occurrence of signal peptide (SP) in all determined differential proteins. By doing so we were able to visualize a small proportion of proteins with a positive prediction of SP (9%).” Could the authors better describe this process? I am not understanding what was predicted or how “positive prediction” was visualized.
Response: Dear reviewer we improved the description as follows:
Due to the lower representation of secreted protein in GO enrichment protocols, we predicted the occurrence of N-terminal signal peptide (SP) in the sequences of all determined differential proteins by the SignalP 6.0 platform(https://services.healthtech.dtu.dk/services/SignalP-6.0/). By doing so we were able to visualize a small proportion of proteins with a positive prediction of SP (9%), most of them associated with secreted proteins by the endomembrane system.
- Figure 4B: Is this Venn diagram for all samples or a specific PSL comparison?
Response: yes, the Venn diagram is related to the differential proteins identified in PSL
- Line 311: (D) should be changed to (E). A description of subfigure 4D is missing.
Response: Many thanks for catching up on several mistakes. We fixed the sentence as follows:
“(D) The percentage of predicted proteins with N-terminal signal peptide (SP) among differential proteins determined in PSL. (E) Heatmap of differential proteins predicted SP”.
- Figure 6B and C: AP, EC and ASG are not defined. AP = anticlinal pegs, EC = epidermal cells, ASG = ?
Response: We added the following specifications. In B and C, AP, anticlinal pegs; EC, epidermal cells; ASG, attached sub-epidermal globules.
- Line 373: Another reference formatting
Response:Done
- Line 505: Remove the author instructions
Response:Done
- SI: There are some formatting errors with units in the SI – specifically there are numerous instances were the μM are missing the “μ”
Response:Done
Major Points:
- The authors should add a short sentence or two at the end of the abstract to describe the outcome of this study – specifically that their data fills a gap in knowledge regarding avocado PSL and the future of the work.
Response: We added the following lines in lines 33-35:
“Our study shed light on the association of proteome modulation with the structural features of Hass avocado. Its detailed characterization will provide an alternative for better preservation during the postharvest period”
- Figure 2D, E and F: The legend scale on all three subfigures should include smaller dots that are closer to the sizes of the dots in the figures. As it is currently presented, readers are required to extrapolate dot sizes smaller to estimate size and count. The legend should span the range of dot sizes included in the figure. This same correction is required for Fig. 4C.
Response: Done